# Dual phase high temperature Si$_3$N$_4$/Al(Ti)N films with tunable thermal conductivity

Zhaohe Gao [1,2,3,8] ✉, Han Liu [1], Jinchi Sun[4], Justyna Kulczyk-Malecka[5], Xiaodong Liu [1], Etienne Bousser[6], Peter Kelly[5], Yu-Lung Chiu [2] ✉, Philip J. Withers [1,7] & Ping Xiao[1] ✉

Engineering amorphous dielectric films with tunable thermal conductivity is advantageous for the thermal management of semiconductor devices and thermal insulation of aerospace applications. Here, we demonstrate that incorporating dense dispersed amorphous Al(Ti)N (~1 nm or above) nanoparticles having phase volume fractions from 6 to 70 %, has a negligible effect on the intrinsic thermal conductivity of the amorphous Si$_3$N$_4$ matrix (~2 W m$^{-1}$K$^{-1}$), in which the wave-like 'propagons' in Allen-Feldmann theory are believed to be unsupressed and non-tuned. By contrast, incorporating (5–15 nm) crystalline TiN phases significantly increases the thermal conductivity (up to 15 W m$^{-1}$K$^{-1}$). Critically, the micrometre-thick Si$_3$N$_4$/AlN and Si$_3$N$_4$/TiN amorphous matrix dual-phase nanocomposite coatings exhibit excellent thermal stability upon exposure to ambient air at 1000 °C for 50 h. These findings shed light on the phonon transport mechanism regarding the effects of the second phase and pave a design pathway for engineering amorphous coatings displaying unprecedented high thermal conductivity and excellent thermal stability.

The thermal conductivity of amorphous dielectric materials plays a significant role in modern semiconductor devices, e.g., gate dielectrics or interlayer dielectrics for metal-oxide-semiconductors, hard mask layers for Si semiconductors, photoelectric conversion layers for solar cells, phase change memory, and in thermal insulation fields, such as high-speed aircraft/spacecraft and for advanced radiation detectors[1-3]. Amorphous dielectric solids have inherently low thermal conductivity, and optimizing heat conduction in these solids is widely acknowledged to be extremely difficult. This is ascribed to the fact that heat conduction in amorphous solids is more complex than for crystalline solids due to their lack of atomic periodicity[4]. Atomic vibrations, whose quanta are known as phonons, serve as the means of heat transport in dielectric materials. In crystalline solids, phonons propagate and scatter, as described by Peierls's formulation of the Boltzmann transport equation[5], having well-defined group velocities. However, for amorphous dielectric solids, the lack of periodicity means that a large portion of the vibration modes do not have a well-defined group velocity. Heat conduction in amorphous dielectric materials has been described by the 'amorphous limit' model originally proposed by Einstein in 1911 and further developed by Slack in 1979 and Cahill in 1992, known as the 'minimum thermal conductivity' model[6,7]. Here the heat conduction is governed by what has subsequently been termed 'diffusons', which are spatially delocalized, exhibiting no apparent periodicity. They carry heat through atomic

¹ Department of Materials, Henry Royce Institute, University of Manchester, Manchester, UK. ²School of Metallurgy and Materials, University of Birmingham, Birmingham, UK. ³Centre for Manufacturing and Materials, Coventry University, Coventry, UK. ⁴Department of Materials Science and Engineering and Materials Research Laboratory, University of Illinois at Urbana-Champaign, Urbana, IL, USA. ⁵Surface Engineering Group, Manchester Fuel Cell Innovation Centre, Manchester Metropolitan University, Manchester, UK. ⁶Department of Engineering Physics, Polytechnique Montréal, Montreal, QC, Canada. ⁷Department of Materials Science and Engineering, Monash university, Clayton, VIC, Australia. ⁸Present address: Materials Genome Institute & State Key Laboratory of Materials for Advanced Nuclear Energy, Shanghai University, Shanghai, China. ✉e-mail: gaozhaohe2013@gmail.com; z_gao@shu.edu.cn; y.chiu@bham.ac.uk; p.xiao@manchester.ac.uk

vibrations by the 'random walk' of the oscillators of the atoms rather than the plane-wave-like propagation typical of crystalline solids. Subsequently, in 1993, Allen and Feldman classified the vibrational modes into three categories: plane-wave-like 'propagons', 'diffusons', and 'locons'[8]. As described by refs. 9,10, propagons have a well-defined wave vector and a periodic form that can produce a propagating wave packet. As low-frequency modes they are delocalized throughout the entire system, while locons are spatially localized modes.

Following Allen and Feldman's theory, propagons contribute about 4% of the total thermal conductivity of amorphous Si, while diffusons and locons contribute about 93% and 3%, respectively[11]. More generally, it has been demonstrated that propagons can make a 13% contribution for amorphous carbon[12], a 30% contribution for amorphous cement[13], a 28–42% contribution for amorphous silicon[14,15], and a 62% contribution for amorphous silicon nitride[4]. Theoretically, wave-like propagons, by analogy with the scattering of phonons, can be suppressed by impurities[16,17], interfaces[18], and surfaces[19,20] in amorphous solids. Unlike impurities or defects, a densely distributed second phase can improve mechanical properties. For example, $Si_3N_4$/MeN (Me = Zr, Al, Ti, Cr, Ta, etc) nanocomposite nitride coatings comprising a stable amorphous $Si_3N_4$ matrix and densely distributed MeN second phases can exhibit enhanced hardness, enhanced resistance to cracking and excellent high temperature stability[21,22]. However, it remains unclear whether a densely dispersed amorphous second phase suppresses the wave-like propagons in the amorphous matrix, while the effects of a densely dispersed crystalline second phase have rarely been reported.

This paper aims to fill these gaps. Specifically, nanocomposite nitride coatings consisting of an amorphous $Si_3N_4$ matrix and amorphous/crystalline AlN/TiN phases deposited by reactive magnetron sputtering are studied to evaluate their thermal performance and to uncover the key phonon transport mechanisms. In this respect, it is noteworthy that phase segregation of SiN and MeN phases (Me = Zr, Al, Ti, Cr, Ta, etc) in the ternary Si–Me–N systems occurs during coating deposition[22,23]. As a result, the volume fractions of the dispersed second phases or the embedding phases can be tailored by controlling the deposition power of the sputtering targets. Furthermore, the amorphous or crystalline state of the dispersed second phase and the embedding phase can be modified by changing the deposition parameters, e.g., the substrate temperature (a high temperature promoting crystalline phases), the deposition power of targets (to modify the Si content), the partial pressure inside chamber, bias voltage, etc[22,24]. In this way, we have been able to fabricate a systematic set (see Table 1) of coatings containing varying volume fractions of amorphous (a-AlN or

a-TiN), or crystalline (c-TiN) particles in an amorphous $a\text{-}Si_3N_4$ matrix pointing the way to a new family of amorphous coatings having adjustable thermal conductivity.

## Results

### Microstructural characterisation

The microstructure and elemental distribution of the as-deposited 15% a-AlN and 61 and 88% c-TiN coatings are shown in Fig.1 (microstructures for the other $a\text{-}Si_3N_4$/a-Al(Ti)N and $a\text{-}Si_3N_4$/c-TiN film systems can be viewed in supplementary Figs. s1-s4).

The 15%a-AlN ($a\text{-}Si_3N_4$/a-AlN) coating appears to be fully dense without any signs of porosity or cracking. The elemental distributions of Si, Al, and N are also uniform, as shown in Fig.1b. The ~300 nm thick Mo interlayer acts as a bonding layer between the coating and the underlying TiAl alloy or Si wafer substrates. The selective area diffraction (SAD) pattern (Fig.1a inset) confirms that this coating is amorphous. It comprises a matrix of $a\text{-}Si_3N_4$ containing a homogeneously distributed array of ~1.1 nm a-AlN particles, as shown in the STEM BF image (Fig.1c inset) obtained by aberration-corrected TEM and also illustrated in supplementary Fig. s5. The amorphous nature of the 31% a-AlN, 6% a-TiN and 70% a-TiN coatings has been confirmed by SAD analysis as well as by XRD analysis (supplementary Figs. s1–s4).

The 88% c-TiN ($a\text{-}Si_3N_4$/c-TiN) coatings have been confirmed by HRTEM (Fig. 1d) to comprise amorphous $Si_3N_4$ containing isolated TiN nanocrystals, along with partially connected TiN nanocrystals, as illustrated in Fig. 1d and confirmed by XRD analysis in supplementary Fig. s4. The 81% c-TiN (e) and 61% c-TiN (f) coatings display the crystalline TiN and amorphous $Si_3N_4$, where, despite the high volume fraction of TiN, the $Si_3N_4$ largely isolates the TiN nanocrystals from one another, as illustrated in Fig. 1e, f, respectively.

### Thermal conductivity

Figure 2 compares the thermal conductivity ($\kappa$) of our amorphous matrix/dispersed amorphous and crystalline phase films with benchmarks from the literature. The $a\text{-}Si_3N_4$ coating displays a low thermal conductivity (1.9 W m$^{-1}$K$^{-1}$) in line with those for $a\text{-}SiN_x$ films from the established literature. The slight variations in the reported thermal conductivity of different $SiN_x$ films could be attributed to different measuring methodologies or interfacial thermal resistance, which can be significant for film thicknesses of one hundred nanometres or below[25,26]. It seems that the contribution of the propagating vibrational modes (propagons) to the thermal conductivity of the amorphous films is not significantly affected by the presence of amorphous densely dispersed a-AlN (a-AlN $\kappa$~1.65 W m$^{-1}$K$^{-1}$ and c- AlN $\kappa$~320 W

**Table 1 | The phase compositions and volume fractions of the investigated $Si_3N_4$/Al(Ti)N coatings as determined by Super-X-EDS in TEM and focused ion FIB-XPS**

| Sample type (composition in at%) | Name | Thickness (μm) | Substrate heat or not | TiN or AlN volume(%) | $Si_3N_4$ volume (%) | Thermal conductivity (W m$^{-1}$K$^{-1}$) (Percentage STDV.P) |
|---|---|---|---|---|---|---|
| **$a\text{-}Si_3N_4$** | a-SiN | 4.8 | no | 0 | 100 | 1.9 (4%) |
| **$a\text{-}Si_3N_4$ + a-AlN systems** | | | | | | |
| Si: 40 ± 1.2, N: 52, Al: 8 ± 0.7 | 15%a-AlN | 1.7 | no | 15 ± 3 | 85 ± 3 | 1.9 (4%) |
| Si:36 ± 1.3, N:45, Al:19 ± 1.1 | 31%a-AlN | 4.6 | no | 31 ± 4 | 69 ± 4 | 2.3 (6%) |
| **$a\text{-}Si_3N_4$ + a-TiN systems** | | | | | | |
| Si:47 ± 1.4, N:50, Ti:3 ± 0.5 | 6%a-TiN | 5.5 | | 6 ± 2 | 94 ± 2 | 2.0 (2%) |
| Si:14 ± 0.5,N:44.7,Ti:41.3 ± 1.2 | 70%a-TiN | 2.1 | no | 70 ± 4 | 30 ± 4 | 2.4 (0.7%) |
| **$a\text{-}Si_3N_4$ + c-TiN systems** | | | | | | |
| Si:17.5 ± 1.1,N:42.9,Ti:39.6 ± 0.9 | 61%c-TiN | 1.0 | 450 °C | 61 ± 4 | 39 ± 4 | 6.4 (5%) |
| Si:7.2 ± 0.8, N:42.6, Ti:50.2 ± 1.3 | 81%c-TiN | 1.4/7.7 | no | 81 ± 5 | 19 ± 5 | 6.9 (2%) |
| Si:4.4 ± 0.4, N:41.5, Ti:54.1 ± 1.7 | 88%c-TiN | 0.8 | no | 88 ± 4 | 12 ± 4 | 15.1 (7%) |
| **c-TiN** | c-TiN | 0.7 | no | 100 | 0 | 32 (13%) |

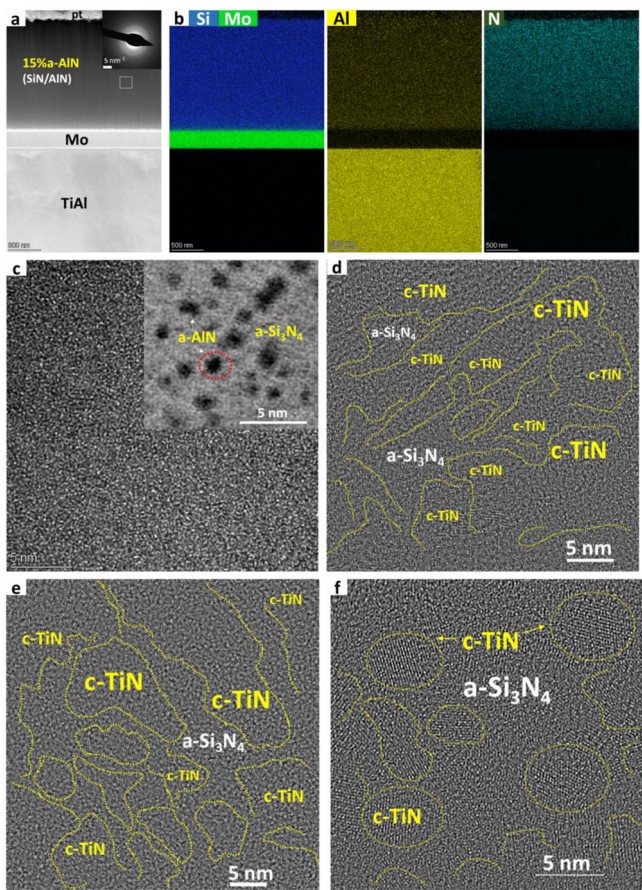

**Fig. 1 | Microstructure and elemental distribution of 15% a-AlN (a-Si₃N₄/a-AlN) and 61 and 88% c-TiN (a-Si₃N₄/c-TiN) coatings. a** Cross-sectional HAADF image of 15% a-AlN coating with Mo interlayer on TiAl alloy, inset showing the SAD pattern acquired from the rectangular box; **b** EDS elemental maps corresponding to the image in (**a**); **c** HRTEM image of 15% a-AlN coating with inset STEM BF image of the 15% a-AlN coating, the dark clusters represent AlN phases; **d** HRTEM image of 88%c-TiN coating showing the crystalline TiN and amorphous Si₃N₄ showing isolated TiN nanocrystals along with partially connected TiN nanocrystals; HRTEM images of **e** 81% c-TiN and **f** 61% c-TiN coatings indicating that despite the high volume fraction of TiN, the Si₃N₄ isolates the TiN nanocrystals from one another.

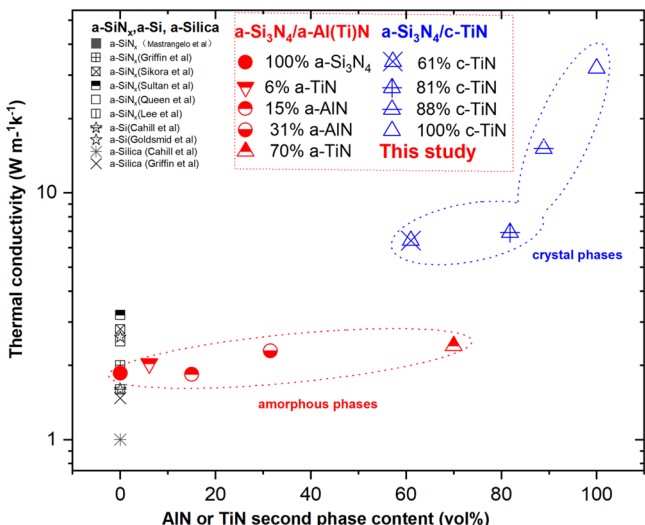

**Fig. 2 | Thermal conductivity as a function of nanoparticle content.** Our a-Si₃N₄/a-Al(Ti)N and a-Si₃N₄/c-TiN films (indicated by circles and triangles respectively) are plotted alongside comparable a-SiNₓ, a-Si, a-silica films from the literature[25,26,50–58].

m⁻¹K⁻¹ [1,27–29]) or a-TiN (a-TiN κ usually below 3 W m⁻¹K⁻¹) second phases since they show similar conductivities to the monolithic a-Si₃N₄ coating. By contrast, the inclusion of c-TiN phases (κ~ 32W m⁻¹K⁻¹), in 61% c-TiN, 81% c-TiN and 88%c-TiN increases the thermal conductivity significantly to 6.4 W m⁻¹K⁻¹, 6.9 W m⁻¹K⁻¹ and 15.1 W m⁻¹K⁻¹, respectively. It is noteworthy that the 88% c-TiN coating shows a significant increase in thermal conductivity in comparison with 81 or 61 % c-TiN. This could be caused by percolation effects associated with particle connectivity of the relatively high thermal conductivity crystalline TiN phase illustrated in Fig.1d. The 61% c-TiN shows 6.4 W m⁻¹K⁻¹ in thermal conductivity, while 81% c-TiN with a relatively high volume of crystalline TiN phases has a similar value of 6.9 W m⁻¹K⁻¹. This surprising outcome could be ascribed to the volume and geometry of crystalline particles, which also affects the thermal conductivity of such nanocomposites. In this respect, the 61% c-TiN coating displays relatively spherical crystalline TiN phases as the substrate temperature rose during coating deposition. This provides a design pathway for a new family of coatings displaying high thermal conductivity.

## Thermal stability
The 15% a-AlN and 81% c-TiN coatings have been selected as representative examples of the respective (a-Si₃N₄/a-AlN) and (a-Si₃N₄/c-

TiN) systems to investigate their thermal stability. Upon exposing them to high temperature in air, most of the amorphous material crystallises. With respect to the 15% a-AlN coating, Fig. 3a and b shows the microstructure and elemental distribution after thermal exposure to 900 °C for 100 h and to 1000 °C for 50 h in air, respectively. After exposure to 900 °C, the 15% a-AlN/Mo interlayer-coated TiAl surface is smooth and free from spallation or cracking, in contrast to the rough surface and severe oxidation exhibited by the bare TiAl alloy surface, as shown in Fig. 3a and supplementary Figs. s6-s8. An (~1 μm thick) oxide layer is evident in Fig. 3a-ii while the interfacial region of the 15% a-AlN coating along with the Mo interlayer shows evidence of an interdiffusion/reaction with the underlying TiAl substrate, as shown in Fig.3a ii and iv, also in supplementary Fig. s6. Nevertheless, the (~1 μm thick) remainder of the 15% a-AlN coating, remains un-oxidised retaining an amorphous microstructure, confirmed by HRTEM and SAD analysis in Fig. 3a ii and iii. By comparison, upon exposure to air at 1000 °C for 50 h, oxidation and interfacial depletion has caused the 15% a-AlN coating to reduce to 300 nm thick, which retains the amorphous state, confirmed by HRTEM analysis in Fig.3b i to iii and shown in supplementary Fig. s7. These results demonstrate that the amorphous 15% a-AlN coating provides good oxidation protection and can withstand high temperature exposure over long durations without undergoing crystallisation.

As for the thermal stability of the amorphous Si₃N₄ systems containing the c-TiN; the 81% c-TiN (a-Si₃N₄/c-TiN, as-deposited thickness 7.7 μm) coating displays good oxidation resistance in air at 900 °C for 50 h giving rise to an oxide scale about 6-7 μm thick, as shown in supplementary Fig. s9. This is somewhat inferior to that for the amorphous dual phase 15% a-AlN (a-Si₃N₄/a-AlN) coating, which gave an ~1 μm oxide scale after 100 h at 900 °C.

These observations are due to the intrinsic thermal stability of the Si₃N₄, which has a high activation energy and an extremely low parabolic rate constant for oxidation[21,30]. This is despite the relatively low activation energies and high parabolic rate constants of oxidation for the dispersed AlN or TiN phases. Also, AlN-containing films are expected to perform better than TiN-containing films due to the protective nature of aluminium oxide[31–33]. As a result, these nanocomposite systems offer the prospect of combining high toughness and hardness[22] with good thermal stability. This is in contrast to low thermal conductivity a-Si or a-Si-H films which tend to undergo crystallisation or oxidation at relatively low temperatures[1,34], or amorphous

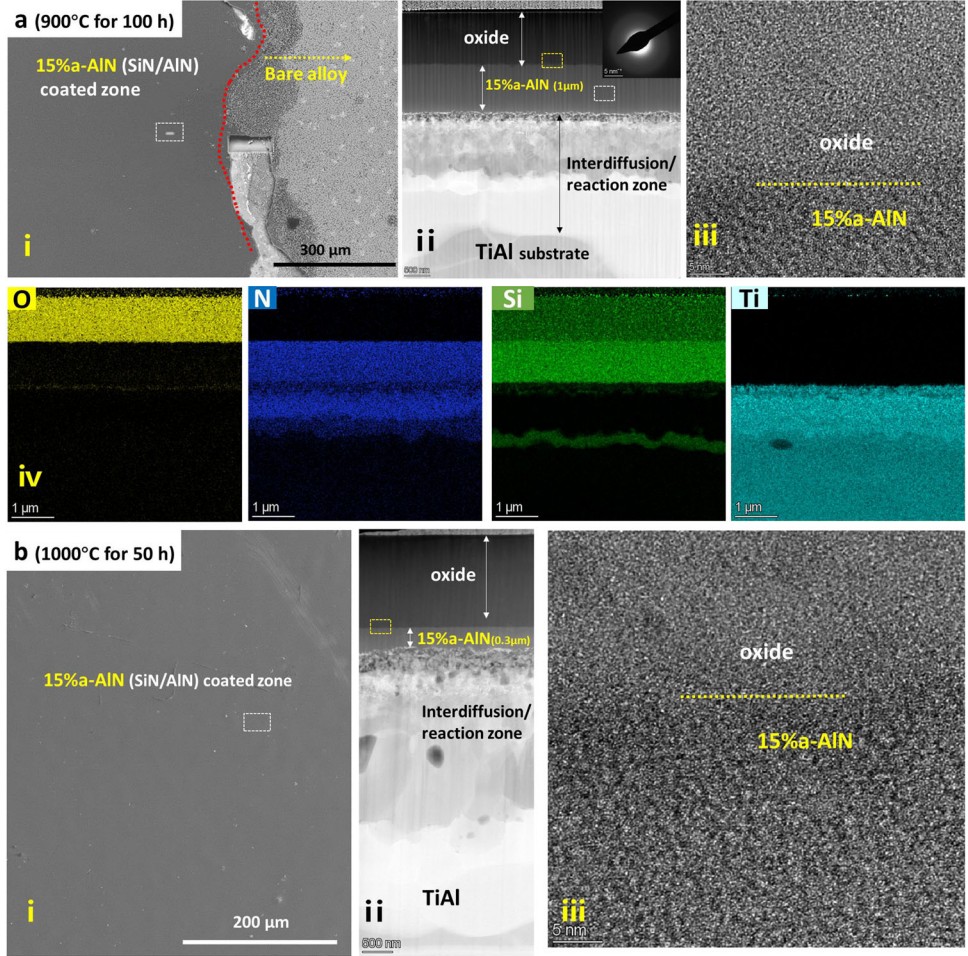

**Fig. 3 | Microstructure of 15% a-AlN coatings after high temperature exposure.** Microstructure and elemental distribution of 15% a-AlN (a-Si$_3$N$_4$/a-AlN) coating **a** after thermal exposure at 900 °C for 100 h and **b** after thermal exposure at 1000 °C for 50 h. In each case (i) shows a plan view SEM image of the top surface of the 15% a-AlN/Mo coated TiAl and bare TiAl alloy; (ii) a cross-sectional HAADF image with SAD from un-oxidised 15% a-AlN for TEM lamellae acquired from the rectangle in (i); and iii) a HRTEM image of the oxidised/unoxidised 15% a-AlN boundary obtained from the yellow rectangle in (ii). iv) shows a series of elemental EDS maps for the image indicated in (ii).

SiN$_x$ or silica/alumina films displaying good high-temperature stability, but which tend to crack easily, especially when deposited on metallic substrates, due to their inherent low toughness relative to the dual phase nanocomposite Si$_3$N$_4$/Al(Ti)N films proposed here. This suggests that the strategy of embedding a-AlN or c-TiN phases into amorphous Si$_3$N$_4$ offers the prospect of coatings for which the thermal conductivity can be tailored with good thermal stability and toughness.

## Discussion

Our results in Fig. 2 suggest that embedding a dense dispersed amorphous Al(Ti)N second phase within an amorphous matrix (see schematic diagrams in Fig.4a−c) does not reduce the thermal conductivity of Si$_3$N$_4$ (~2 W m$^{-1}$K$^{-1}$) significantly, even at 70% volume fraction. These values are well above the (~1 W m$^{-1}$K$^{-1}$) limit widely reported from the 'minimum thermal conductivity' classical model[7,9,35]. This suggests that for these amorphous dual-phase nitrides, the plane-wave-like 'propagons' proposed by Allen and Feldman contribute to the thermal transport.

It has previously been proposed that wave propagation in amorphous solids can be tuned and suppressed[18]. For example, for amorphous Si$_3$N$_4$ containing a square array of air channels of diameter 25 nm and 81 nm, the measured intrinsic thermal conductivity could be decreased from 2.7 W m$^{-1}$K$^{-1}$ (a-Si$_3$N$_4$ film without air channels) to about 1.1 and 0.9 W m$^{-1}$K$^{-1}$, respectively. Phonon wave interference and suppression effects occur at periodic interfaces formed by cylindrical channels inside a film, e.g., a periodically porous a-Si$_3$N$_4$ film incorporating a square array of cylindrical air pores, or at periodic interfaces formed by a multi-layer coating system, e.g., a-Si/a-SiO$_2$ layered superlattice structures[4,18,35–37]. It is important to note that the periodicity dominates the phonon wave interference, thereby suppressing wave propagation. Moreover, the highly smooth interfaces (above-mentioned periodic interface, low interface roughness) and longer mean free paths for phonons could promote wave interference effects[18,38–40]. However, in our case, the 1 nm diameter dispersed AlN second phase is discontinuous, fine and approximately spherical in shape, but does not display any periodicity in terms of its distribution within the amorphous matrix, as illustrated in Fig.4b, c. Taken together, the lack of periodicity of the AlN phase along with the intrinsic short mean free path in the a-S$_3$iN$_4$ film (tens to hundreds of nanometres[4]) means that the chance of propagating wave interference happening is low. The interfacial thermal resistance (diffuse scattering) between the second phase AlN and S$_3$iN$_4$ is determined by their Debye temperature ratio[41]. As a consequence, their extremely close Debye temperatures[42] give a relatively low interfacial thermal resistance. Thus, in the absence of periodicity, the nanoscale second-phase particles likely can not suppress the plane-wave-like propagation and thus reduce their thermal conductivity.

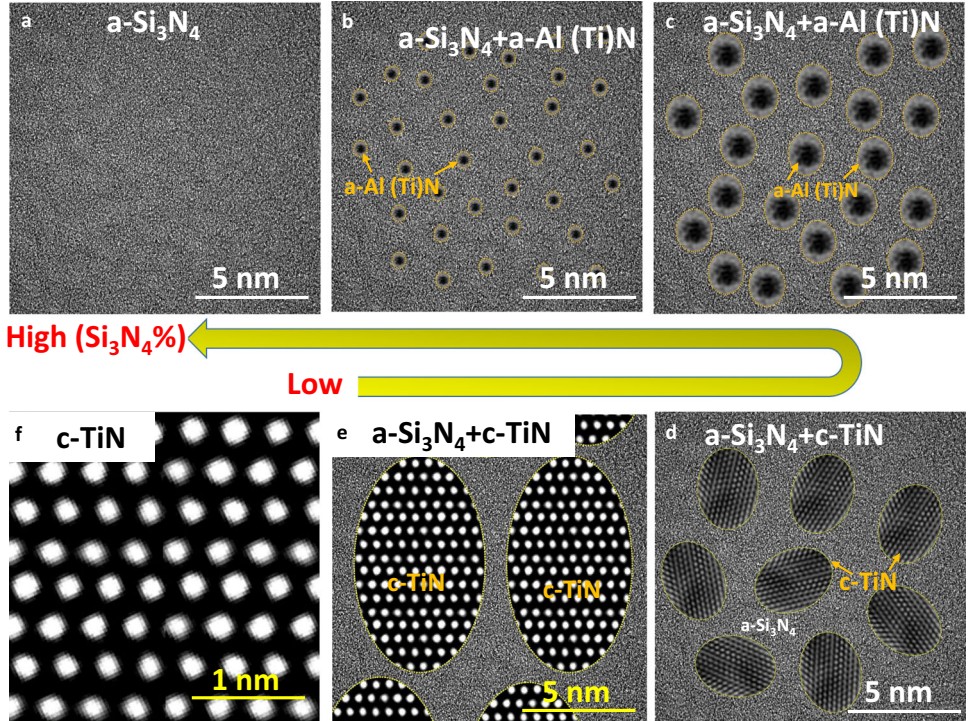

**Fig. 4 | Schematic diagrams of dual-phase nanocomposite Si₃N₄/Al(Ti)N films as a function of Si₃N₄ content.** Idealised diagrams drawn on the basis of our TEM images. **a** 100% a-Si₃N₄; **b** a-Si₃N₄ matrix with a dispersed 1-2 nm diameter a-Al(Ti)N phase, **c** a-Si₃N₄ matrix with a dispersed 2–5 nm or above a-Al(Ti)N phase, **d** a-Si₃N₄ matrix containing dispersed 5–10 nm c-TiN phase; **e** a minor amount of a-Si₃N₄ separating a high volume fraction of 10 nm or above c-TiN phase thereby forming a nanocomposite; **f** a 100% c-TiN coating. The 61 vol% c-TiN sample is quite consistent with the structure depicted in (d), while (e) resembles the microstructure of 81% c-TiN samples. Phase segregation of Si₃N₄ and MeN phases (Me = Zr, Al, Ti, Cr, Ta, etc) in the ternary Si−Me−N systems occurs during coating deposition, which could be modified by changing the deposition parameters, e.g., Si power, substrate temperature, bias voltage, or partial pressure inside the chamber.

By contrast with the low thermal conductivity of amorphous dual phase a-Si₃N₄/a-Al(Ti)N nitrides, embedding c-TiN phases (61% c-TiN, 81% c-TiN, and 88% c-TiN) as illustrated in Fig.4d, e give rise to high thermal conductivities up to 15.1 W m⁻¹K⁻¹ (in Fig. 2), which is 8 times higher than that of the amorphous 15% a-AlN coating. Typically, such an amorphous (a-Si₃N₄)/crystalline (c-TiN) nanocomposite coating could be regarded as a mixture of a low thermal conductivity phase and a high thermal conductivity phase. This could be modelled by the Maxwell-Garnett Effective Medium Approximation (MG-EMA) in which the interface effect and spherical particle size are included[43], or by the Eshelby inclusion method[44]. Embedding a relatively high thermal conductivity material, such as crystalline TiN (32 W m⁻¹K⁻¹), provides a means of tailoring the intrinsically low thermal conductivity of amorphous Si₃N₄ (2 W m⁻¹K⁻¹) by modifying the volume fraction of crystalline TiN. Indeed, the c-TiN (32 W m⁻¹K⁻¹) could be replaced by even higher thermal conductivity crystalline phases, e.g., c-BN (-1000 W m⁻¹K⁻¹), θ-TaN (-995 W m⁻¹K⁻¹), or c-AlN (-320 W m⁻¹K⁻¹)[28,45,46], likely leading to a-Si₃N₄/c-B(Ta/Al)N nitrides with thermal conductivities in excess of 100 W m⁻¹K⁻¹. As a result, strengthening c-MeN (Me = Zr, Al, Ti, Cr, Ta, B, etc.) nanoparticles could be tailored to optimise both the mechanical properties and thermal conductivity of a-Si₃N₄ coatings simultaneously. Moreover, SiN/AlN coating along with Mo interlayer on Ti alloys has displayed good adhesion and conformability upon thermal cycling testing[21], and thereby it could be potentially applied in aerospace industries.

In conclusion, the thermal conductivities of Si₃N₄/AlN and Si₃N₄/TiN nanocomposite coatings comprising an amorphous (a) Si₃N₄ matrix and an amorphous (a) or crystalline (c) dispersed phase have been studied. The following conclusions can be drawn:

I.  Incorporating ~1 nm or above amorphous Al(Ti)N at volume fractions from 6–70 vol% did not significantly affect the thermal conductivity. This is contrary to the simple 'minimum thermal conductivity' classical model. It is suggested that this is because the amorphous AlN or TiN phase, lacking periodicity in its distribution, does not suppress the wave-like 'Propagons' (in the Allen-Feldmann theory) in the amorphous Si₃N₄ matrix.

II. The addition of increasing levels (from 61−88 vol%) of isolated crystalline TiN nanoparticles gave rise to a sharp increase in the thermal conductivity of the film.

III. Although the films containing a high fraction of c-TiN performed least well, the addition of both amorphous AlN and crystalline TiN phases gave rise to micrometre-thick films that are stable and provided good oxidation resistance at 900 °C for 100 h and at 1000 °C for 50 h.

This work provides a design pathway for a new family of amorphous matrix coatings by which we can not only tailor their thermal conductivity, but also achieve good thermal stability and mechanical properties.

## Methods

### Coating process

The Si₃N₄/Al(Ti)N coatings were deposited on one side of commercial (50 × 50 × 0.6 mm³) Si wafers for thermal conduction testing and on polished (50 × 30 × 3 mm³) TiAl alloy coupons for high temperature thermal exposure testing. The TiAl plates had previously been ground, polished (surface roughness, Ra 60 nm) and ultrasonically pre-cleaned in acetone. Deposition was carried out using reactive sputtering in a Teer Coatings' magnetron sputtering system, as detailed elsewhere[30]. In all cases, prior to the deposition of the Si-based layer, a thin (about 300 nm) molybdenum interlayer was deposited in an argon-only atmosphere (-2.1 × 10⁻³ mbar, 0.21 Pa) to provide good interfacial

bonding with the underlying substrate. Three unbalanced (300 × 100 mm³) magnetrons were installed around a rotating unheated substrate holder. Depending on the coating composition, the magnetrons were fitted with 99.5% pure Si, Ti, Al, or Mo targets. Before deposition, the chamber was evacuated to a base pressure below $1 \times 10^{-3}$ Pa. The substrates were sputter cleaned by Ar⁺ ions at a bias voltage of 600 V DC for 15 min prior to the deposition. The Si, Ti, Al, and Mo targets were powered by Advanced Energy Pinnacle Plus power supplies operating in pulsed DC mode at powers ranging from 0 to 1000 W. And the power density of targets could be calculated by power/geometry of the target. The total power of Si and Al was quite similar to that of Si and Ti. Considering the plasma heating up holder effect, the substrate temperature shall be the same. Heating up the sample holder with an extra heating source would almost certainly promote the crystalisation of TiN or AlN. The 61%vol c-TiN samples have been deposited with the heating-up samples holder (setting up at 450 °C). And, high deposition temperature could promote the crystallization of TiN phase. The pulse frequency was 100 kHz with a duty cycle of 60%, a DC bias of −30 V being applied to the substrate during coating deposition. The compositions of the SiN, TiN, SiAlN and SiTiN coatings were modified by applying different powers to the targets. The deposition parameters have been described in more detail previously[21,47]. The film thicknesses and phase fractions are summarised in Table 1 using the naming convention where 15%a-AlN represents a film containing 15 vol% of amorphous AlN in an a-Si₃N₄ matrix. We have used the elemental ratio of Si, Ti and the corresponding cell volume size of the corresponding nitrides to calculate the volume percentage of c-TiN and a-Si₃N₄ or a-AlN. The TiN-Si₃N₄ phase diagram did not show any stable ternary phases for TiSiN and Ti has a low chance to incorporate into a-SiN, vice versa. Thus, the volume calculation of TiN did not consider the elemental incorporation effects. In this work, we aimed to study the effect of different volume percentages of c-TiN or a-SiN. The incorporated effect should be identical and should be negligible. Moreover, the cell volume expansion effect of amorphous Si₃N₄ has also been considered.

### Microstructure and compositional characterisation

The compositions of the as-deposited Si₃N₄/Al(Ti)N coatings were analysed by focused ion beam X-ray photoelectron spectroscopy (FIB-XPS, Kratos AXIS Supra) along with transmission electron microscopy (on a TEM, FEI Talos F200A fitted with the Super-X-EDS system). To observe the microstructure and composition of the as-deposited coatings and oxidized samples in greater detail, thin lamellae of the cross-sections of the coatings were prepared by focused ion beam (FIB, FEI Helios 660, Helios G5) using the lift-out technique and then examined by high-resolution transmission electron microscopy (HRTEM) and scanning transmission electron microscopy (STEM) (TEM, FEI Talos F200A). Atomically resolved STEM images were acquired using an aberration-corrected dedicated scanning transmission electron microscope (FEI, Titan G2) equipped with high-, middle-, and low-angle annular dark-field (HAADF, MAADF, and LAADF) and annular bright-field (ABF) STEM detectors. Over 30 HRTEM or STEM images of each sample have been checked to accurately faithfully characterise the nanocomposite microstructures.

### Thermal conductivity and high temperature stability testing

Time-domain thermoreflectance (TDTR), an ultra-fast laser-based pump-probe metrology setup in the Professor Cahill group at the University of Illinois at Urbana-Champaign, was used to measure thermal conductivity of the as-deposited Si₃N₄/Al(Ti)N coatings. An approx. 80 nm thick Al film is first coated on the sample by magnetron sputtering to serve as the transducer. The pump and probe laser beams are focused on the Al film by a 5x objective lens, resulting in beam radii of 9.7 μm. The intensity modulated pump laser heats the surface of the sample periodically with a modulation frequency of

9.7 MHz. The probe laser detects the temperature variation at the surface of the sample by thermoreflectance, i.e., the change of optical reflectivity with temperature, see refs. 48,49. The ratio signal as a function of time is fitted by an analytical model with the thermal conductivity of the nitride and the thermal boundary conductance of the interface between Al and nitride as the fitting parameters.

In order to assess the thermal stability and ability to resist oxidation in high-temperature air, we conducted tests on samples (3 to 5 samples for each condition) of SiN/AlN-coated TiAl alloy and SiN/TiN in a furnace under static air conditions. The targeted working temperature of nitride coatings on Ti alloy or TiAl alloys for aero engine blades shall be 600–800 °C. The testing temperature, 900 °C, is quite harsh and 1000 °C could be regarded as extremely harsh. The samples were placed inside the furnace as soon as it was turned on. The ramp rate was 10 °C/min and the testing period began when the temperature stabilized at the desired level. After the targeted testing period, the furnace was powered off and the samples were taken out when the temperature was below 50 °C.

## Data availability

The authors declare that the data supporting the findings of this study are available from the corresponding authors upon request.

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

## Acknowledgements

Z. Gao is grateful to Prof. David G. Cahill at the University of Illinois at Urbana-Champaign for his valuable suggestions and guidance in this work, and his help in TDTR measurement. Z. Gao is also grateful to Prof. Felix Hofmann and Dr. Abdallah Reza at Oxford University for his help in thermal conductivity measurement via TGS. Z. Gao is grateful to Dr. Xun Zhang from the University of Manchester for his valuable advice and inspiration on heat transfer and H. Liu for her perseverance in film thermal measurement via Laser Flash. Z. Gao and H. Liu are also grateful to Dr. Alexander Carruthers from Manchester University for his help in HRTEM image processing on phase segregation. PJW is grateful to Monash University for hosting him during his sabbatical. PJW and PX are grateful to the Henry Royce Institute for Advanced Materials, established through EPSRC grants EP/R00661X/1, EP/P025498/1, and EP/P025021/1. PX acknowledges the Royal Academy of Engineering and Rolls-Royce for his Rolls-Royce/Royal Academy of Engineering Research Chair in Advanced Coating Technology.

## Author contributions

Z.Gao proposed and designed the project, conceived the idea for the work, and drafted this work while affiliated with the University of Manchester and University of Birmingham. H.L., X.L. and Z.G. carried out the TEM operation and the analyses. J.S. contributed to the thermal conductivity measurement via TDTR. J.K.-M., E.B., P.K. and Z.G. fabricated the coating samples. Y.C., P.W., and P.X. refined this work and also provided supportive resources to help to finish this work. All authors provided critical feedback and helped shape the manuscript.

## Competing interests

The authors declare no competing interests.
