## [Transparent Peer Review file · Nature Communications]

Dual phase high temperature Si₃N₄/Al(Ti)N films with tunable thermal conductivity

Corresponding Author: Professor Ping Xiao

Version 0:

Reviewer comments:

Reviewer #1

(Remarks to the Author)

Paper summary:

The authors investigate thermal conductivity in nanocomposite films that consists of an amorphous Si₃N₄ matrix with various amounts of nanoparticle fillers in the range of 6 – 92 vol%. They study amorphous nanoparticle fillers (a-TiN or a-AlN) in the range of 6 – 70 vol%. They also study crystalline nanoparticle fillers (c-TiN) in the range of 88-92 vol%. I have some major concerns about this paper as described below:

Comments.

1. Achieving an amorphous matrix nanocomposite (92 vol% crystalline particles that are isolated from one another by 8 vol% amorphous material) with a 15 W/m-K thermal conductivity would be quite impressive. However, it is not convincing they have a sample like this (see comments 2-6 below)
2. If one of the two phases in a dual-phase nanocomposite is 92 vol%, it is highly unlikely that that phase consists of nanoparticles that are truly isolated from one another via amorphous material. In all likelihood, that 92 vol% phase is continuous and highly percolated throughout the sample (i.e. it is the “matrix” of the nanocomposite). If the 92 vol% crystalline phase is continuous and highly-percolated, then a 15 W/m-K thermal conductivity is no longer impressive.
3. There is very limited materials characterization of their samples that are claimed to be 88 vol% and/or 92 vol% crystalline TiN nanoparticles. One characterization is included as Figure 1d and is a TEM image of the 92 vol% c-TiN sample. However, from my view of this image, it looks like they have much less than 92 volume % of the crystalline phase.
4. Do the authors have any other materials characterization that confirms their sample composition and morphology? Figure 1d is insufficient for multiple reasons. First (as mentioned in comment 2 above), that figure doesn't not appear to show 92 volume %. Second, Figure 1d is a single TEM image of a very small portion of a single sample. Additional characterization methods (including those that can sample large global portions of the sample) are needed.
5. A simple exercise that supports the difficulty of achieving the sample described by the authors is the consideration of closed-packed-particles of spherical shape and uniform size. In this case, the maximum achievable vol% would be 74% (i.e. FCC packing of particles that “touch” one another). Hence all of their crystalline nanoparticle films are quite a bit greater than this limit and that makes me skeptical of their claim. Admittedly, the nanoparticle fillers of the authors don't appear to be spherical (Fig 1d), however, the difficulty of achieving that 92 vol% of isolated particles are demonstrated by this simple example.
6. The vast majorities of characterizations appear to be done on the 15 vol% amorphous AlN samples. However, it is the 88 and 92 vol% crystalline TiN samples that are the most interesting in this work. Why is there so little characterization of those samples?

7. There are no error/uncertainty bars throughout their work. This error/uncertainty analysis needs to be done.
8. There are no statistics in their work (or at least these aren't made clear). How many samples were made and measured when it came to each sample type?
9. Figure 4 is somewhat mis-leading has the figure "resembles" actual microscopy images (i.e. includes precise scale bars, with black/white contrast color scheme, and even has "noise" within the schematics). While people who do electron microscopy can see that these aren't "real" images, this might not be obvious to many readers (or to students reading this paper). I suggest that this figure be more obviously labeled as being not real. Perhaps remove the scale bars and maybe use a color-scheme that is not black/white.
10. The Figure 4d, 4e "schematics" don't represent the claimed 88-92 vol% of crystalline TiN filler that the authors claim is in their sample (see line 202 on Page 10). Rather, based upon my eyeball viewing, it is drawn to be quite a bit less than the high volume % that is claimed. This figure should be revised accordingly.
11. The authors call their samples "amorphous coatings" throughout the paper which is misleading. Their high conductivity sample contains 92 vol% of crystalline nanoparticle fillers (and is hence just 8 vol% amorphous). This is vastly more crystalline than it is amorphous. Hence the language here is misleading. I suggest the authors be more precise with their language. Calling these "amorphous matrix composites" or something similar is probably suitable (assuming that convincing data can be produced that illustrates that the nanoparticles are isolated from one another - see comments 2-6).

Reviewer #2

(Remarks to the Author)

This is a well-executed study on $\text{Si}_3\text{N}_4/\text{Al}(\text{Ti})\text{N}$ coatings with tunable thermal conductivity and good thermal stability. The experimental data are solid, and the topic is of high relevance. However, key assumptions—such as propagons' behavior, interfacial effects, and matrix continuity at high TiN content—require further justification. With improved mechanistic clarity and application-level evaluation, this work could make a valuable contribution to the field.

1. The authors state that even at a high volume fraction (~70%) of amorphous $\text{Al}(\text{Ti})\text{N}$, the thermal conductivity remains largely unchanged, suggesting that propagons remain unaffected. However, is this supported by any quantitative modeling? Can TDTR measurements alone sufficiently support this assumption?
2. Given that most films are 1–5 μm thick, TDTR measurements are sensitive to interfacial thermal resistance—especially for multilayered or composite films. The manuscript does not provide data or correction methods for interfacial resistance. Could this lead to overestimated thermal conductivity?
3. The authors state that the 15% a-AIN film remains amorphous after 1000°C annealing. Is this conclusion based solely on SAD and HRTEM, or is there a quantification of the crystallization extent?
4. The manuscript asserts that even in the 92% c-TiN sample, Si_3N_4 still isolates the TiN nanocrystals. However, given the high volume fraction of TiN, is it accurate to describe this system as Si_3N_4 -based? Does the TEM evidence support the presence of a percolating matrix phase?
5. The authors suggest potential applications in aerospace and semiconductor industries. However, have they evaluated the coatings' mechanical integrity, adhesion strength, or electrical insulation properties under device-level conditions?
6. In the 88% and 92% c-TiN samples, thermal conductivity rises sharply. Besides the high intrinsic conductivity of TiN, could size effects, percolation, or interconnected grain networks be influencing heat transport, rather than composition alone?

Reviewer #3

(Remarks to the Author)

This is original research paper in characterizing thermal conductivity and thermal stability properties of $\text{Si}_3\text{N}_4/\text{TiN}$ and $\text{Si}_3\text{N}_4/\text{AlN}$ nanocomposite coatings. Overall the methodology is solid, the results are novel and well presented, and there interesting discussions about influence of amorphous and crystalline inclusions of AlN and TiN on the thermal conductivity in amorphous Si_3N_4 matrix as well as on the oxidation resiliency of such composite films at high temperature exposure.

I have few points where I believe some clarity may help to the paper readers:

1. Would it be possible to add to Figure 2 some thermal conductivity data from literature for AlN and TiN films with a different degree of crystallinity (few nm to about micron)? If such data can be found, it would be helpful to compare with the reported in this paper measurements for composite films. I am aware of at least one study where AlTiN thermal conductivity was investigated as a function of grain size crystallinity and cubic to wurtzite transformation during high temperature annealing (Richard Rachbauer et al., *Acta Materialia* 60 (2012) 2091–2096). That study shows a clear dependence of thermal conductivity on grain sizes (reduces at small sizes and increases at larger sizes). Perhaps there are other studies of thermal conductivity as a function of grain sizes, as both TiN and AlN films were produced for few decades.
2. The crystalline TiN is a reasonably good thermal conductor, and for 88-92% of c-TiN inclusions, thermal transport could be dictated by TiN phase with scattering at grain boundaries and between grains in Si_3N_4 . In such, the mechanism of thermal transport in such nanocomposite is likely a mix of modes dictated by both amorphous matrix and crystalline grains depending on their relative volume fractions and interfaces. Perhaps include a comment on such in the discussion.

3. What was deposition temperature during film growth? While the substrate holder was not heated, the Ar plasma cleaning step and the magnetron plasma are likely to heat the deposition surface. TiN inclusions were formed as crystalline, while AlN inclusions were grown as amorphous. Is this due to the substrate temperature induced by a difference of powers on Ti and Al sputtering targets? Substrate temperature and/or target power density during the film growth could be added to methodology section.

4. In thermal stability tests: what is more harsh – holding for longer time at 900C (100 hours in the study), or holding for shorter time at 1000C (50 hours in the study)? The latter is probably harsher, but a clarification comment on such would be helpful to help differentiate between the tests, since both temperature and time were varied.

5. In Figure 3. The Mo EDS map is missing. The Mo from the interface layer is likely to diffuse.

6. In terms of ability to resist high temperature oxidation, the AlN containing films were well expected to perform better due to the protective nature of aluminum oxide. There are multiple reports on such high temperature oxidation protection with Al-containing hard coatings (AlTiN, AlTiVN, etc.), e.g. tailored for hard coatings on cutting tools for dry machining operations. Adding few references to such papers in the discussion can help to enhance the point of using AlN inclusions toward a higher thermal stability.

7. On lines 150-151 "...good oxidation resistance in air at 900C for 50 hours..." – should this be for 100 hours? Looks like a typo.

Version 1:

Reviewer comments:

Reviewer #1

(Remarks to the Author)

Paper summary:

The paper appears to be improved, but still has shortcomings as described below. The new data point for 64% c-TiN also opens additional questions about interpretation of their thermal conductivity data. It is worth noting that my comments 3-4 below are critical of the quoted values for the volume percent of crystalline TiN content and this continues on from my prior review. With that being said, it would seem that the crystalline volume percent is lower than they claim... If that's the case, the thermal conductivity values they measure become more noteworthy due to decreased crystallinity relative to their quoted values. I do find the thermal conductivity values quite interesting, but I still have some main concerns as described below.

Comments.

1. The thermal transport results are quite intriguing. The new 64% c-TiN sample is intriguing because it is comparable to the 71% a-TiN sample, but has a >3x higher thermal conductivity. This sample is also interesting because somehow it has a comparable thermal conductivity to the 88% c-TiN sample. Why should both 64% c-TiN and 88% c-TiN have comparable thermal conductivities of ~6.4 – 6.9 W/m-K, but then a small increase up to 92% c-TiN double in value up to 15.1 W/m-K? I suspect percolation effects and particle connectivity is playing a role in the 92% sample. This needs to be explained better.

2. The 64% c-TiN and 88% c-TiN have should have meaningfully different amounts of amorphous material (i.e. the amorphous material content in the 88% sample should be 1/3 that of the 64% sample). Hence, these samples should have meaningfully different thermal conductivities. However, these thermal conductivities are very similar (6.4 vs 6.9 W/m-K). This is a strange result that merits explanation. Are the amorphous materials in these two samples somehow different from one another?

3. I understand and appreciate the efforts that the authors made to confirm their volume percent of c-TiN. However, this remains an ongoing main concern I have about the manuscript. It's not clear to me how they would get a sample that is 88 volume percent crystalline TiN particles that are distinct and separated by amorphous a-Si₃N₄. The microscopy images simply do not appear to be 88 volume percent of TiN particles as claimed. I've attached a figure to this review to make my point. This attached figure shows side-by-side (a) the TEM image of the authors 88 vol% sample (Figure 1e in manuscript) and (b) a scaled drawing I made of what an 88 vol% sample would look like. Side-by-side inspection of the attached figure shows the quoted volume percent and the microscopy images don't agree with one another.

4. My understanding is that they used the elemental ratio of Si:Ti and the corresponding cell volume size of the corresponding nitrides. Could these lead to errors? (a) Is it possible for the titanium to alloy into the a-Si₃N₄ matrix and/or the silicon to alloy into the TiN particles? (b) (a) What is the uncertainty of the elemental content measurement? Could these or other reasons explain why the microscopy images and the quoted volume percentages don't seem to agree with one another?

Reviewer #2

(Remarks to the Author)

All questions raised have been addressed and resolved.

Reviewer #3

(Remarks to the Author)

Thank you for revising the manuscript in according to provided comments. I do not have any additional comments. This is a very thorough and detailed study. My recommendation is to accept the revised version.

Version 2:

Reviewer comments:

Reviewer #1

(Remarks to the Author)

The authors have adequately addressed my comments. I support publication of this latest version of the manuscript.

REVIEWER COMMENTS

We are grateful to the reviewers for their comments. We have responded to their comments in blue type. The associated changes to the paper are marked with yellow highlighter.

Reviewer #1 (Remarks to the Author):

Paper summary:

The authors investigate thermal conductivity in nanocomposite films that consists of an amorphous Si_3N_4 matrix with various amounts of nanoparticle fillers in the range of 6 – 92 vol%. They study amorphous nanoparticle fillers (a-TiN or a-AlN) in the range of 6 – 70 vol%. They also study crystalline nanoparticle fillers (c-TiN) in the range of 88-92 vol%. I have some major concerns about this paper as described below:

Comments.

1. Achieving an amorphous matrix nanocomposite (92 vol% crystalline particles that are isolated from one another by 8 vol% amorphous material) with a 15 W/m-K thermal conductivity would be quite impressive. However, it is not convincing they have a sample like this (see comments 2-6 below)

Reply: We pick up this point in response to comments 2-6 below.

2. If one of the two phases in a dual-phase nanocomposite is 92 vol%, it is highly unlikely that that phase consists of nanoparticles that are truly isolated from one another via amorphous material. In all likelihood, that 92 vol% phase is continuous and highly percolated throughout the sample (i.e. it is the “matrix” of the nanocomposite). If the 92 vol% crystalline phase is continuous and highly-percolated, then a 15 W/m-K thermal conductivity is no longer impressive.

Reply: We understand the concern from the reviewer. However, this work uncovers the dual phase concept of a- Si_3N_4 /c-TiN films with tunable thermal conductivity. Of course, the likelihood of c-TiN being isolated by a- Si_3N_4 would increase as function of the increase of Si_3N_4 . We have re-checked the HRTEM images of 92%vol c-TiN sample and also added HRTEM images of 88%vol and 64%vol c-TiN samples, to Fig.1 in the revised manuscript.

The 92%vol c-TiN sample displays isolated TiN nanocrystals along with partially connected TiN nanocrystals after checking above 30 HRTEM images. The 88%vol c-TiN sample displays isolated TiN nanocrystals. To get accurate results we have developed a new method based on python codes to process the HRTEM images and identify the crystal zone and amorphous zone of such nanocomposites. The HRTEM images and processed images for 92% and 88% c-TiN are displayed below in Fig.α. The crystal zone (c-TiN) should show relative bright in this image.

The 64% vol c-TiN sample has been newly added to the revised manuscript. It is evident that the c-TiN nanocrystals have been fully isolated by amorphous Si_3N_4 , illustrated in Fig.1f.

We have re-examined the literature relating to a-SiN/c-TiN nanocomposites. The Si content, partial pressure during deposition, the bias voltage, the substrate temperature all affect whether c-TiN particles are fully isolated or not [Reference 22-24 in the manuscript]. The benchmark point for Si content, ~7 to 13%, and above this value it easily forms nanocomposite [1,3] [Reference 22-24 in manuscript]. Fig.β in the literature has displayed one type of such nanocomposite[1]. The volume percentage of nitrides investigated here was calculated based on the atomic contents of Si, Ti and the cell volume size of corresponding nitrides. Such a-SiN/c-TiN, a-SiN/c-CrN, a-SiN/c-AlN, etc. nanocomposites definitely are achievable. Moreover, SiTiN samples with 7-8 at %Si content [2] (corresponding to our 88%vol c-TiN sample) have been found to display a fully isolated c-TiN particle morphology.

As for the value of thermal conductivity, $6.4 \text{ W m}^{-1}\text{K}^{-1}$ for 64% vol c-TiN and $15.1 \text{ W m}^{-1}\text{K}^{-1}$ for 92% vol c-TiN are far higher than about 2 W/mK for all amorphous cases. In fact, the c-TiN ($32 \text{ W m}^{-1}\text{K}^{-1}$) could be replaced by even higher thermal conductivity crystalline phases, e.g. c-BN ($\sim 1000 \text{ W m}^{-1}\text{K}^{-1}$), θ -TaN ($\sim 995 \text{ W m}^{-1}\text{K}^{-1}$) or c-AlN ($\sim 320 \text{ W m}^{-1}\text{K}^{-1}$) potentially leading to a-Si₃N₄/c-B(Ta/Al)N nitrides with thermal conductivities in excess of $100 \text{ W m}^{-1}\text{K}^{-1}$.

Fig.1 Microstructure and elemental distribution of 15%a-AlN (a-Si₃N₄/a-AlN) and 64 to 92%c-

TiN (a-Si₃N₄/c-TiN) coatings. (a) Cross-sectional HAADF image of 15%a-AlN coating with Mo interlayer on TiAl alloy, inset showing the SAD pattern acquired from the rectangular box; (b) EDS elemental maps corresponding to the image in (a); (c) HRTEM image of 15%a-AlN coating with inset STEM BF image of the 15%a-AlN coating, the dark clusters represent AlN phases; (d) HRTEM image of 92%c-TiN coating showing the crystalline TiN and amorphous Si₃N₄ showing isolated TiN nanocrystals along with partially connected TiN nanocrystals; HRTEM images of (e) 88%c-TiN and (f) 64%c-TiN coatings indicating that despite the high volume fraction of TiN, the Si₃N₄ isolates the TiN nanocrystals from one another.

Fig.α The HRTEM images and python processed images of 88% and 92%c-TiN (a-Si₃N₄/c-TiN) coatings. A 128x128 pixel sliding box was placed over the images, and translated in steps of 4 pixels. At each location of the sliding box, the fft of the region was calculated, and the maximum within a radius of 10-20 pixels from the centre of the fft was found. Where crystallinity is present, the fft show a bright spot in this region. Some of the images show a few erroneous bright spots. This is likely just the result of random noise within ffts occasionally generating a bright pixel.

[Figure Redacted]

Fig.β Schematic map of the microstructural evolution in the Cr-Si-N films as function of Si content and negative bias voltage. [reprocessed from [1]]

3. There is very limited materials characterization of their samples that are claimed to be 88 vol% and/or 92 vol% crystalline TiN nanoparticles. One characterization is included as Figure 1d and is a TEM image of the 92 vol% c-TiN sample. However, from my view of this image, it looks like they have much less than 92 volume % of the crystalline phase.

Reply: This is a good question. We have re-prepared TEM samples of 88 vol% and 92% c-TiN samples. And over 30 HRTEM images of each sample have been checked. The corresponding update has been added into revised manuscript, Page 12. The revised HRTEM images of 88 vol% and 92% c-TiN have been updated in the revised manuscript to be more representative. The volume percentage was calculated based on the atomic contents of Si, Ti and the cell volume size of corresponding nitrides.

4. Do the authors have any other materials characterization that confirms their sample composition and morphology? Figure 1d is insufficient for multiple reasons. First (as mentioned in comment 2 above), that figure doesn't appear to show 92 volume %. Second, Figure 1d is a single TEM image of a very small portion of a single sample. Additional characterization methods (including those that can sample large global portions of the sample) are needed.

Reply: This was an important point and we have looked again at our results and tried to demonstrate this more clearly in the revised text. We have undertaken analysis via STEM-EDS, FIB-XPS. In addition, HRTEM and XRD have been used to investigate the phases and microstructure of 92% c-TiN samples. About 30 HRTEM images of 92% c-TiN sample have been checked. A more representative HRTEM image of 92% c-TiN has been added to the revised manuscript. Moreover, the HRTEM images of 88%vol and 64%vol c-TiN samples have also been added into revised manuscript. In summary we have undertaken a systematic microstructural evolution of dual phases.

5. A simple exercise that supports the difficulty of achieving the sample described by the authors is the consideration of closed-packed-particles of spherical shape and uniform size. In this case, the maximum achievable vol% would be 74% (i.e. FCC packing of particles that "touch" one another). Hence all of their crystalline nanoparticle films are quite a bit greater than this limit and that makes me skeptical of their claim. Admittedly, the nanoparticle fillers of the authors don't appear to be spherical (Fig 1d), however, the difficulty of achieving that 92 vol% of isolated particles are demonstrated by this simple example.

Reply: We understand the concern from the reviewer. The microstructure of 92%vol c-TiN, 88%vol c-TiN and the newly added 64%vol c-TiN samples have been displayed and discussed above. This work aims to give a systematic description of microstructural evolution across the increasing TiN sample set. Definitely, 64%vol c-TiN in the Fig.1f displays the isolated TiN particles. The Si content, partial pressure during deposition, the bias voltage, the substrate temperature all affect whether c-TiN particles are fully isolated or not. This has been discussed above and also in manuscript. The benchmark point for Si content, ~7 to 13%, and above this value it easily forms nanocomposite[1,3] [Reference 22-24 in manuscript]. Moreover, M. Bartosik have reported that in their case the TiN crystallites were encapsulated by an ~1.0 nm thick SiN tissue phase and this would easily form nanocomposite at relative low content of Si[3].

For our investigated study, Si content is 17.5at% for 64% c-TiN sample, 7.2at% for 88% c-TiN sample. And HRTEM analysis of both samples have displayed isolated TiN nanocrystals.

6. The vast majorities of characterizations appear to be done on the 15 vol% amorphous AlN samples. However, it is the 88 and 92 vol% crystalline TiN samples that are the most interesting in this work. Why is there so little characterization of those

samples?

Reply: We have taken this point on board and now now added HRTEM analysis of a 64vol% c-TiN sample to complement those for 88, 92 vol% c-TiN samples. The thermal stability performance of 88vol% c-TiN have added and is illustrated in supplementary Fig.s9.

7. There are no error/uncertainty bars throughout their work. This error/uncertainty analysis needs to be done.

Reply: We apologise as we should have included error bars. The percentage STDV.P for all measured thermal conductivity has been added into revised Table 1.

8. There are no statistics in their work (or at least these aren't made clear). How many samples were made and measured when it came to each sample type?

Reply: Again, we are pleased to clarify. We now mention that around 30 TEM micrographs were examined. In addition between 3 to 5 samples were examined for each sample type. This point has been added to the revised manuscript.

9. Figure 4 is somewhat mis-leading has the figure "resembles" actual microscopy images (i.e. includes precise scale bars, with black/white contrast color scheme, and even has "noise" within the schematics). While people who do electron microscopy can see that these aren't "real" images, this might not be obvious to many readers (or to students reading this paper). I suggest that this figure be more obviously labeled as being not real. Perhaps remove the scale bars and maybe use a color-scheme that is not black/white.

Reply: Thanks reviewer for the good suggestion. We have re-drawn this schematic diagram with colored lines, illustrated below and also updated in revised manuscript.

Fig.4 Schematic diagrams of dual phase nanocomposite $\text{Si}_3\text{N}_4/\text{Al}(\text{Ti})\text{N}$ films as function of different Si_3N_4 content. Idealised diagrams drawn on the basis of our TEM images. (a) a- Si_3N_4 ; (b) a- Si_3N_4 matrix with a dispersed 1-2 nm diameter a-Al(Ti)N phase, (c) a- Si_3N_4 matrix with a

dispersed 2-5 nm or above a-Al(Ti)N phase, (d) a-Si₃N₄ matrix containing dispersed 5-10 nm c-TiN phase; (e) a minor amount of a-Si₃N₄ separating a high volume fraction of 10 nm or above c-TiN phase thereby forming a nanocomposite; (f) a c-TiN coating. The 64vol% c-TiN sample is quite consistent with structure depicted in Fig.4 d. Fig.4 e resembles the microstructure of 88% c-TiN samples. Phase segregation of Si₃N₄ and MeN phases (Me=Zr, Al, Ti, Cr, Ta, etc) in the ternary Si–Me–N systems occurs during coating deposition, which could be modified by changing the deposition parameters, e.g. Si power, substrate temperature, bias voltage or partial pressure inside chamber.

10. The Figure 4d, 4e “schematics” don’t represent the claimed 88-92 vol% of crystalline TiN filler that the authors claim is in their sample (see line 202 on Page 10). Rather, based upon my eyeball viewing, it is drawn to be quite a bit less than the high volume % that is claimed. This figure should be revised accordingly.

Reply: To respond to this good point we have added the 64vol% c-TiN sample which is quite consistent with structure depicted in Fig.4 d. Similarly, the schematic diagram in Fig.4 e resembles the microstructure of 88% c-TiN samples. This has been added into the caption of Figure 4 in the revised manuscript, Page 9.

11. The authors call their samples “amorphous coatings” throughout the paper which is misleading. Their high conductivity sample contains 92 vol% of crystalline nanoparticle fillers (and is hence just 8 vol% amorphous). This is vastly more crystalline than it is amorphous. Hence the language here is misleading. I suggest the authors be more precise with their language. Calling these “amorphous matrix composites” or something similar is probably suitable (assuming that convincing data can be produced that illustrates that the nanoparticles are isolated from one another - see comments 2-6).

Reply: We are pleased to pick up this point. In the revised manuscript, we have updated it as amorphous matrix nanocomposite.

Reviewer #2 (Remarks to the Author):

This is a well-executed study on Si₃N₄/Al(Ti)N coatings with tunable thermal conductivity and good thermal stability. The experimental data are solid, and the topic is of high relevance. However, key assumptions-such as propagons’ behavior, interfacial effects, and matrix continuity at high TiN content-require further justification. With improved mechanistic clarity and application-level evaluation, this work could make a valuable contribution to the field.

Reply: We would like to express our sincere gratitude to the reviewer for the recognition of our work.

1. The authors state that even at a high volume fraction (~70%) of amorphous Al(Ti)N, the thermal conductivity remains largely unchanged, suggesting that propagons remain unaffected. However, is this supported by any quantitative modeling? Can TDTR measurements alone sufficiently support this assumption?

Reply: TDTR measurements give the real value of thermal conductivity for the investigated nitride nanocomposites as this method is an accurate one with a very limited range of errors.

In practice, modeling can also be useful in predicting thermal conductivity. We have used Molecular Dynamics to model such an amorphous nanocomposite. However, it is challenging to model the thermal conductivity of these nanocomposites because of the amorphous status and randomly distributed second phases.

2. Given that most films are 1–5 μm thick, TDTR measurements are sensitive to interfacial thermal resistance—especially for multilayered or composite films. The manuscript does not provide data or correction methods for interfacial resistance. Could this lead to overestimated thermal conductivity?

Reply: The thermal penetration depth in the TDTR measurement is much smaller than the thickness of the investigated films. As the result, the sample behaves effectively as a bulk material and the thickness does not matter. Besides, in the analysis of the TDTR signal, the thermal conductivity of the material and the thermal boundary conductance (Al transducer/nitride coating) are fitted together. The thermal conductivity is mostly sensitive to the absolute value of the ratio signal while the thermal boundary conductance is sensitive to the slope of the ratio signal versus the delay time. This fitting approach works well for the measured samples as validated by the good repeatability of measured values.

3. The authors state that the 15% α -AlN film remains amorphous after 1000°C annealing. Is this conclusion based solely on SAD and HRTEM, or is there a quantification of the crystallization extent?

Reply: The 15% α -AlN film remains amorphous after 1000°C annealing. This is based on SAD and HRTEM images. Also, the remnant 15% α -AlN still remains amorphous without crystallization based on the HRTEM images of 5 locations. This has been updated '15% α -AlN coating, remains un-oxidised retaining an amorphous microstructure, confirmed HRTEM and SAD analysis..' in the revised manuscript, Page 7.

4. The manuscript asserts that even in the 92% c-TiN sample, Si_3N_4 still isolates the TiN nanocrystals. However, given the high volume fraction of TiN, is it accurate to describe this system as Si_3N_4 -based? Does the TEM evidence support the presence of a percolating matrix phase?

Reply: About 30 HRTEM images analysis of 88, 92 vol% c-TiN samples and newly added 64vol%TiN sample have been examined. Representative images have been added into revised manuscript, illustrated in Fig.1.

As we have said in response to reviewer 1 (question 2), the 92%vol c-TiN sample displays isolated TiN particles along with partially connected TiN particles after checking above 30 HRTEM images. The 88%vol c-TiN sample displays isolated TiN particles. The 64% vol c-TiN sample is newly added into revised manuscript. It is quite obvious that the c-TiN nanoparticles have been fully isolated by amorphous Si_3N_4 .

5. The authors suggest potential applications in aerospace and semiconductor industries. However, have they evaluated the coatings' mechanical integrity, adhesion strength, or electrical insulation properties under device-level conditions?

Reply: Thanks for making this point. Actually, the adhesion strength of such coatings has been evaluated in our previous work [4]. SiAlN/Mo coating on Ti alloys has displayed good adhesion and conformability upon thermal cycling testing. In the future, we will investigate the electrical insulation properties under device-level conditions.

In the revised manuscript, the sentence 'Moreover, SiN/AlN coating along with Mo interlayer on Ti alloys has displayed good adhesion and conformability upon thermal cycling testing²¹, and thereby it could be potentially applied in aerospace industries.' has been added, Page 10.

6. In the 88% and 92% c-TiN samples, thermal conductivity rises sharply. Besides the high intrinsic conductivity of TiN, could size effects, percolation, or interconnected grain networks be influencing heat transport, rather than composition alone?

Reply: Thanks reviewer for the good suggestion. This work aims to uncover the effects of dual phase on thermal conductivity. The size and geometry of particles shall affect the thermal

conductivity of such nanocomposites. The difficulty is how to accurately achieve the expected size and distribution of TiN particles during coating deposition, illustrated in below schematic diagram. In the future, we aim to investigate the thermal conductivity affected by size and distribution of TiN.

Fig.y Schematic diagrams of SiN/Ti(Al)N and Ti(Al)N phases displaying different geometry.

Reviewer #3 (Remarks to the Author):

This is original research paper in characterizing thermal conductivity and thermal stability properties of Si₃N₄/TiN and Si₃N₄/AlN nanocomposite coatings. Overall the methodology is solid, the results are novel and well presented, and there interesting discussions about influence of amorphous and crystalline inclusions of AlN and TiN on the thermal conductivity in amorphous Si₃N₄ matrix as well as on the oxidation resiliency of such composite films at high temperature exposure.

Reply: We would like to express our sincere gratitude to the reviewer for the recognition of our work.

I have few points where I believe some clarity may help to the paper readers:

1. Would it be possible to add to Figure 2 some thermal conductivity data from literature for AlN and TiN films with a different degree of crystallinity (few nm to about micron)? If such data can be found, it would be helpful to compare with the reported in this paper measurements for composite films. I am aware of at least one study where AlTiN thermal conductivity was investigated as a function of grain size crystallinity and cubic to wurtzite transformation during high temperature annealing (Richard Rachbauer et al., Acta Materialia 60 (2012) 2091–2096). That study shows a clear dependence of thermal conductivity on grain sizes (reduces at small sizes and increases at larger sizes). Perhaps there are other studies of thermal conductivity as a function of grain sizes, as both TiN and AlN films were produced for few decades.

Reply: The crystallinity and grain size definitely affect the thermal conductivity of crystal AlN or TiN films. For the recommended AlTiN work, it focuses on phase transition and corresponding thermal conductivity evolution of the monolithic single phase cubic Ti_{1-x}Al_xN into cubic TiN and wurtzite AlN.

The recommended reference has been cited in the revised manuscript, Page 5.

Our work focuses on the dual phase film system in which amorphous SiN_x matrix and amorphous or crystal AlN or TiN second phases. Purely grain size effect of single crystal AlN or TiN was not been focused. But, the spherical particle size of AlN or TiN has been considered by the Maxwell-Garnett Effective Medium Approximation and the Eshellby inclusion method

in the manuscript.

2. The crystalline TiN is a reasonably good thermal conductor, and for 88-92% of c-TiN inclusions, thermal transport could be dictated by TiN phase with scattering at grain boundaries and between grains in Si₃N₄. In such, the mechanism of thermal transport in such nanocomposite is likely a mix of modes dictated by both amorphous matrix and crystalline grains depending on their relative volume fractions and interfaces. Perhaps include a comment on such in the discussion.

Reply: The Maxwell-Garnett Effective Medium Approximation and Eshelby inclusion method have been used to discuss the thermal conductivity of mixed phases. In the manuscript, the sentence 'Typically, such an amorphous (a-Si₃N₄)/crystalline (c-TiN) nanocomposite coating could be regarded as a mixture of a low thermal conductivity phase and high thermal conductivity phase. This could be modelled by the Maxwell-Garnett Effective Medium Approximation (MG-EMA) in which the interface effect and spherical particle size are included, also could be modelled by Eshelby inclusion method' has been illustrated.

3. What was deposition temperature during film growth? While the substrate holder was not heated, the Ar plasma cleaning step and the magnetron plasma are likely to heat the deposition surface. TiN inclusions were formed as crystalline, while AlN inclusions were grown as amorphous. Is this due to the substrate temperature induced by a difference of powers on Ti and Al sputtering targets? Substrate temperature and/or target power density during the film growth could be added to methodology section.

Reply: The temperature around sample holder could reach about 200-300°C during coating deposition because of the plasma. The power density could be calculated by power/target geometry (300 × 100 mm²). In the revised manuscript, the power density and depositing temperature have been added in Table 1 in revised manuscript. The total power of Si and Al was quite similar to that of Si and Ti. Thus, the substrate temperature shall be same.

Heating up the sample holder with an extra heating source would almost certainly promote the crystallisation of TiN or AlN. The 64%vol c-TiN samples have been newly deposited with the heating-up samples holder (setting up at 450°C). And, high deposition temperature could promote the crystallisation of TiN phase.

The corresponding updates have been added into revised manuscript, Page 11.

4. In thermal stability tests: what is more harsh – holding for longer time at 900C (100 hours in the study), or holding for shorter time at 1000C (50 hours in the study)? The latter is probably harsher, but a clarification comment on such would be helpful to help differentiate between the tests, since both temperature and time were varied.

Reply: The 900°C for 100 h is quite harsh for such coating on TiAl alloys. The targeted working temperature of nitride coatings on Ti alloy or TiAl alloys for aero engine blades shall be 600-800°C. The testing temperature, 900°C, is quite harsh and 1000°C could be regarded as extreme harsh.

'The targeted working temperature of nitride coatings on Ti alloy or TiAl alloys for aero engine blades shall be 600-800°C. The testing temperature, 900°C, is quite harsh and 1000°C could be regarded as extreme harsh.' has been added into the revised manuscript, Page 13.

5. In Figure 3. The Mo EDS map is missing. The Mo from the interface layer is likely to diffuse.

Reply: The Mo EDS map has been illustrated in the supplementary Fig.s6 and is referred to in the main text.

6. In terms of ability to resist high temperature oxidation, the AlN containing films were well expected to perform better due to the protective nature of aluminum oxide. There

are multiple reports on such high temperature oxidation protection with Al-containing hard coatings (AlTiN, AlTiVN, etc.), e.g. tailored for hard coatings on cutting tools for dry machining operations. Adding few references to such papers in the discussion can help to enhance the point of using AlN inclusions toward a higher thermal stability.

Reply: Thanks reviewer for the good suggestion. In the revised manuscript, the corresponding references have been added.

7. On lines 150-151 "...good oxidation resistance in air at 900C for 50 hours..." – should this be for 100 hours? Looks like a typo.

Reply: Thanks for pointing this out, the testing duration for 88%c-TiN was 50 h and corresponding characterization after thermal exposure was illustrated in supplementary Fig.s9.

References

[1] Qi Min, etc. Microstructural control of Cr–Si–N films by a hybrid arc ion plating and magnetron sputtering process. *Acta Materialia* 57 (2009) 14974-4987.

[2] Nina Schalk, etc. Nanocomposite versus solid solution formation in the TiSiN system. *Acta Materialia* 275 (2024) 120063.

[3] M. Sperr, etc. Correlating elemental distribution with mechanical properties of TiN/SiNx nanocomposite coatings. *Scripta Materialia* 170 (2019) 20–23.

[4] Z. Gao, etc. A conformable high temperature nitride coating for Ti alloys, *Acta Materialia* 189 (2020) 274-283.

REVIEWER COMMENTS

We are grateful to the reviewers for their comments. We have responded to their comments in blue type. The associated changes to the paper are marked with yellow highlighter.

Reviewer #1 (Remarks to the Author):

Paper summary:

The paper appears to be improved, but still has shortcomings as described below. The new data point for 64% c-TiN also opens additional questions about interpretation of their thermal conductivity data. It is worth noting that my comments 3-4 below are critical of the quoted values for the volume percent of crystalline TiN content and this continues on from my prior review. With that being said, it would seem that the crystalline volume percent is lower than they claim. If that's the case, the thermal conductivity values they measure become more noteworthy due to decreased crystallinity relative to their quoted values. I do find the thermal conductivity values quite interesting, but I still have some main concerns as described below.

Reply: We are grateful to the reviewer for his/her consideration on our work which has stimulated us to go back and re-examine some of our results.

Comments.

1. The thermal transport results are quite intriguing. The new 64% c-TiN sample is intriguing because it is comparable to the 71% a-TiN sample, but has a >3x higher thermal conductivity. This sample is also interesting because somehow it has a comparable thermal conductivity to the 88% c-TiN sample. Why should both 64% c-TiN and 88% c-TiN have comparable thermal conductivities of $\sim 6.4 - 6.9 \text{ W m}^{-1}\text{K}^{-1}$, but then a small increase up to 92% c-TiN double in value up to $15.1 \text{ W m}^{-1}\text{K}^{-1}$? I suspect percolation effects and particle connectivity is playing a role in the 92% sample. This needs to be explained better.

Reply: In response to these comments, we have reanalysed the microstructures and compositions (as discussed in answer to point 3) assigned the previous volume fractions (92% c-TiN, 88% c-TiN, 64% c-TiN) have been revised to (88% c-TiN, 81% c-TiN, 61% c-TiN), respectively, illustrated in below Table and in the revised manuscript.

On reflection, we agree with the reviewer that percolation effects and particle connectivity may play a role in the conductivity of the 88% sample. In the revised manuscript, we have added the sentence 'The 88% c-TiN shows a significant increase in thermal conductivity in comparison with 81 or 61 % c-TiN. This could be caused by percolation effects associated with particle connectivity of the relatively high thermal conductivity crystal TiN phase illustrated in Fig.1 d.'

2. The 64% c-TiN and 88% c-TiN have should have meaningfully different amounts of amorphous material (i.e. the amorphous material content in the 88% sample should be 1/3 that of the 64% sample). Hence, these samples should have meaningfully different thermal conductivities. However, these thermal conductivities are very similar (6.4 vs 6.9 W/m-K). This is a strange result that merits explanation. Are the amorphous materials in these two samples somehow different from one another?

Reply: The reviewer makes a good point. The previous volume fractions 64% c-TiN, 88% c-TiN have been corrected as 61% c-TiN, 81% c-TiN, respectively. The 61% c-TiN shows $6.4 \text{ W m}^{-1}\text{K}^{-1}$ in thermal conductivity while 81% c-TiN with a relatively high volume of crystal TiN phases has a similar value of $6.9 \text{ W m}^{-1}\text{K}^{-1}$. This could be ascribed to the influence of the volume and geometry of crystal particles on the thermal conductivity of such nanocomposites. The difficulty is how to accurately control the size and distribution of TiN particles during coating

deposition, illustrated in below schematic diagram. In the future, we aim to investigate the thermal conductivity affected by size and distribution of TiN.

In the revised manuscript we have added, 'The 61% c-TiN shows $6.4 \text{ W m}^{-1}\text{K}^{-1}$ in thermal conductivity while 81% c-TiN with a relatively high volume of crystal TiN phases has a similar value of $6.9 \text{ W m}^{-1}\text{K}^{-1}$. This surprising outcome could be ascribed to that the volume and geometry of crystal particles which also affects the thermal conductivity of such nanocomposites. In this respect, the 61% c-TiN displays relatively spherical crystal TiN phases as the substrate temperature rose during coating deposition.'

Fig.α1 Schematic diagrams of SiN/TiN and TiN phases displaying different geometry.

3. I understand and appreciate the efforts that the authors made to confirm their volume percent of c-TiN. However, this remains an ongoing main concern I have about the manuscript. It's not clear to me how they would get a sample that is 88 volume percent crystalline TiN particles that are distinct and separated by amorphous a-Si₃N₄. The microscopy images simply do not appear to be 88 volume percent of TiN particles as claimed. I've attached a figure to this review to make my point. This attached figure shows side-by-side (a) the TEM image of the authors 88 vol% sample (Figure 1e in manuscript) and (b) a scaled drawing I made of what an 88 vol% sample would look like. Side-by-side inspection of the attached figure shows the quoted volume percent and the microscopy images don't agree with one another.

Reply: We appreciate the concerns from the reviewer and we are grateful for the diagram provided by the reviewer. This is an important point and we have looked again at our results. We have re-checked chemical analysis via STEM-EDS line scanning, FIB-XPS(line scanning). The error on atomic percentage of Si and Ti elements have been updated in the revised manuscript. We have used the elemental ratio of Si, Ti and the corresponding cell volume size of the corresponding nitrides to calculate the volume percentage of c-TiN and a-Si₃N₄. The cell volume expansion effect of amorphous Si₃N₄ has also been considered. As a result, the previous 88%c-TiN has been corrected to 81%c-TiN. The red box occupies 81% and the blue area is 19%. Considering the non-uniform distribution and non-regular shape of c-TiN, the c-TiN zone illustrated by HRTEM image of is quite close to 80%, to certain extent. Moreover, we visually draw the yellow dash line to separate the c-TiN and a-Si₃N₄ and it may cause some errors the extent of which we now express as an uncertainty in our volume fraction estimates.

Fig.α2 Left figure is the HRTEM image of 81%c-TiN (previous 88%)samples. Right figure consists of 81% red area and 19% blue area.

In the revised manuscript, the corresponding compositional errors and volume percentage have been updated in Table 1.

Table 1 The phase compositions and volume fractions of the investigated $\text{Si}_3\text{N}_4/\text{Al}(\text{Ti})\text{N}$ coatings as determined by Super-X-EDS in TEM and focused ion FIB-XPS.

Sample type (composition in at%)	Name	Thickness (μm)	Substrate heat or not	TiN or AlN volume(%)	Si_3N_4 volume (%)	Thermal conductivity ($\text{W m}^{-1}\text{K}^{-1}$) (Percentage STDV.P)
a-Si_3N_4	a-SiN	4.8	no	0	100	1.9 (4%)
a-Si_3N_4 + a-AlN systems Si: 40 ± 1.2 , N: 52, Al: 8 ± 0.7 Si: 36 ± 1.3 , N: 45, Al: 19 ± 1.1	15%a-AlN	1.7	no	15 ± 3	85 ± 3	1.9 (4%)
	31%a-AlN	4.6	no	31 ± 4	69 ± 4	2.3 (6%)
a-Si_3N_4 + a-TiN systems Si: 47 ± 1.4 , N: 50, Ti: 3 ± 0.5 Si: 14 ± 0.5 , N: 44.7, Ti: 41.3 ± 1.2	6%a-TiN	5.5	no	6 ± 2	94 ± 2	2.0 (2%)
	70%a-TiN	2.1	no	70 ± 4	30 ± 4	2.4 (0.7%)
a-Si_3N_4 + c-TiN systems Si: 17.5 ± 1.1 , N: 42.9, Ti: 39.6 ± 0.9 Si: 7.2 ± 0.8 , N: 42.6, Ti: 50.2 ± 1.3 Si: 4.4 ± 0.4 , N: 41.5, Ti: 54.1 ± 1.7	61%c-TiN	1.0	450°C	61 ± 4	39 ± 4	6.4 (5%)
	81%c-TiN	1.4/7.7	no	81 ± 5	19 ± 5	6.9 (2%)
	88%c-TiN	0.8	no	88 ± 4	12 ± 4	15.1 (7%)
c-TiN	c-TiN	0.7	no	100	0	32 (13%)

4. My understanding is that they used the elemental ratio of Si:Ti and the corresponding cell volume size of the corresponding nitrides. Could these lead to errors? (a) Is it possible for the titanium to alloy into the a- Si_3N_4 matrix and/or the silicon to alloy into the TiN particles? (b) (a) What is the uncertainty of the elemental content measurement? Could

these or other reasons explain why the microscopy images and the quoted volume percentages don't seem to agree with one another?

Reply: The understanding of reviewer is correct. We did use the elemental ratio of Si, Ti and the corresponding cell volume size of the corresponding nitrides to calculate the volume percentage of c-TiN and a-Si₃N₄. It did translate to errors or uncertainty due to the measurement uncertainty caused by STEM-EDS line scanning and FIB-XPS(line scanning).In the revised manuscript, we have added the uncertainty of Si, Ti and Al elemental contents.

The reviewer made a good point on whether Si is incorporated into TiN vice versa. The TiSiN coatings have excellent hardness and thermal stability and such excellent properties originate from their nanocomposite structure, consisting of nanocrystalline(nc) TiN phase embedded in an amorphous(a) SiN_x tissue phase [1-2]. This nanocomposite concept was proposed by Veprek et al in 1995 [3]. A large number of papers and patents have reported this nanocomposite during last 30 years.

Since the TiN-Si₃N₄ phase diagram does not show any stable ternary phases for TiSiN, many researchers have found and reported such nanocomposite structure consisting of nc-TiN in an a-Si₃N₄ matrix [4-5]. Our results investigated here also show this nanocomposite structure. However, nc-Ti_{1-x}Si_xN where Si incorporate into crystal TiN and a-Si_{1-y}Ti_yN where Ti is incorporated into amorphous SiN were found by a few studies [5-6]. But, the incorporated content, x or y, is quite low.

For this work, we aim to study the effect of different volume percentage of nc-TiN or a-SiN. The incorporated effect is similar and likely negligible.

To reflect this point, in the revised manuscript, we have described how we calculate the phase volume and also consider the uncertainty. 'We have used the elemental ratio of Si, Ti and the corresponding cell volume size of the corresponding nitrides to calculate the volume percentage of c-TiN and a-Si₃N₄ or a-AlN. The TiN-Si₃N₄ phase diagram did not show any stable ternary phases for TiSiN and Ti has low chance to incorporate into a-SiN vice versa. Thus, the volume calculation of TiN did not consider the elemental incorporation effects. In this work, we aimed to study the effect of different volume percentage of nc-TiN or a-SiN. The incorporated effect should be identical and should be negligible. Moreover, the cell volume expansion effect of amorphous Si₃N₄ has also been considered.'

Reviewer #2 (Remarks to the Author):

All questions raised have been addressed and resolved.

Reply: We are grateful to the reviewer for his or her ratification.

Reviewer #3 (Remarks to the Author):

Thank you for revising the manuscript in according to provided comments. I do not have any additional comments. This is a very thorough and detailed study. My recommendation is to accept the revised version.

Reply: We are grateful to the reviewer for his or her praise.

References

- [1] M. Diserens, J. Patscheider, F. Levy. Mechanical properties and oxidation resistance of nanocomposite TiN-SiN_x physical-vapor-deposited thin films, Surf. Coat. Technol. 120–121 (1999) 158–165.
- [2] J. Musil. Hard nanocomposite coatings: Thermal stability, oxidation resistance and toughness. Surf. Coat. Technol. 207, 50-65 (2012).
- [3] S. Vepřek, S. Reiprich, A concept for the design of novel superhard coatings, Thin Solid Films 268 (1995) 64–71.
- [4] P. Rogl, J.C. Schuster, Phase Diagrams of Ternary Boron Nitride and Silicon Nitride Systems, ASM International, Materials Park, OH, 1992.
- [5] N. Schalk; etc. Nanocomposite versus solid solution formation in the TiSiN system. Acta Mater. 275 (2024) 120063.
- [6] S. Naghdali etc. Investigation of nanocomposite formation in TiSiN coatings using atom probe tomography. Surf. Coat. Technol. 513 (2025) 132535.

On the left of the figure is the TEM of the author's 88 volume percent c-TiN particle sample (Figure . This appears to be much less than 88 volume percent. As a reference drawing, I've made an 88 vol% drawing on the bottom right. The background blue box is 25 cm² and the summation of the red boxes is 22.1 cm², or 88 %. Clearly the fill fractions of these two things look very different. How or why do these fill fractions not look similar?